# Effect of Dose and Timing of Burdock (*Arctium lappa*) Root Intake on Intestinal Microbiota of Mice

**DOI:** 10.3390/microorganisms8020220

**Published:** 2020-02-06

**Authors:** Aya Watanabe, Hiroyuki Sasaki, Hiroki Miyakawa, Yuki Nakayama, Yijin Lyu, Shigenobu Shibata

**Affiliations:** 1Laboratory of Physiology and Pharmacology, School of Advanced Science and Engineering, Waseda University, Shinjuku-ku, Tokyo 162-8480, Japan; aya_watanabe7115@suou.waseda.jp (A.W.); hiroyuki-sasaki@asagi.waseda.jp (H.S.); hgbbst-hiroki@toki.waseda.jp (H.M.); yukibecky-6991@akane.waseda.jp (Y.N.); ikin@fuji.waseda.jp (Y.L.); 2AIST-Waseda University Computational Bio Big-Data Open Innovation Laboratory (CBBD-OIL), National Institute of Advanced Industrial Science and Technology, Shinjuku-ku, Tokyo 169-85555, Japan

**Keywords:** *A. lappa* (*Arctium lappa*), inulin/fructan, intestinal microbiota, chrononutrition

## Abstract

Water-soluble dietary fiber such as inulin improves the beta diversity of the intestinal microbiota of mice fed with a high-fat diet (HFD). The circadian clock is the system that regulates the internal daily rhythm, and it affects the pattern of beta diversity in mouse intestinal microbiota. Burdock (*Arctium lappa*) root contains a high concentration of inulin/fructan (approximately 50%) and is a very popular vegetable in Japan. *Arctium lappa* also contains functional substances that may affect intestinal microbiota, such as polyphenols. We compared the effects of inulin and *A. lappa* powder on the diversity of the intestinal microbiota of HFD-fed mice. 16S rDNA from the intestinal microbiota obtained from feces was analyzed by 16S Metagenomic Sequencing Library Preparation. It was found to have a stronger effect on microbiota than inulin alone, suggesting that inulin has an additive and/or synergic action with other molecules in *A. lappa* root. We examined the effects of intake timing (breakfast or dinner) of *A. lappa* on intestinal microbiota. The intake of *A. lappa* root in the evening had a stronger effect on microbiota diversity in comparison to morning intake. Therefore, it is suggested that habitual consumption of *A. lappa* root in the evening may aid the maintenance of healthy intestinal microbiota.

## 1. Introduction

The mammalian intestine contains approximately 100 trillion bacteria, which are collectively known as the intestinal microbiota. Recently, a number of studies have examined intestinal microbiota using next-generation sequencing, and many relationships have been found between the intestinal microbiota and physical and pathological conditions [1,2,3]. It is well established that water-soluble dietary fibers provide nutrition for microbiota in the large intestine. Among soluble dietary fibers, inulins have been well researched [4,5]. Inulins can be synthesized by chemical reactions, but they can also be extracted from vegetables [6,7]. There are many plants and vegetables that contain a high concentration of inulin, such as Jerusalem artichoke and chicory [4,5]. The root of *Arctium lappa*, known as Burdock or bardane, is a very popular cultivated edible vegetable in many countries. Interestingly, *A. lappa* root containing a high concentration of inulin is a very popular vegetable in Japan; there are many Japanese recipes that include *A. lappa*, and there are many opportunities to eat *A. lappa* rather than Jerusalem artichoke or chicory. Therefore, in the present experiments, we focused on *A. lappa* root rather than other inulin/fructan-rich vegetables such as artichoke or chicory.

It has been reported that extracts from different parts of *A. lappa* have different biological activity, such as anti-oxidant [8], anti-cancer [9,10], anti-diabetic [11], anti-tubercular [12], anti-inflammatory [13], anti-bacterial, and anti-viral effects [14]. Studies have confirmed that *A. lappa* contains various bioactive molecules, such as polyphenols including caffeic acid derivatives and flavonoids, oligosaccharides, and polyunsaturated fatty acids [15]. Recently, an increasing number of studies have focused on polysaccharides extracted from *A. lappa*, especially fructans, which were used as an inulin source [8,16]. Pectin was one of the non-inulin polysaccharides that was extracted from the root of *A. lappa*, and it was found to show significant anti-constipation activity [17].

Therefore, our investigations focus on examining whether the consumption of *A. lappa* root results in any change in the pattern of microbiota in comparison to inulin, because *A. lappa* contains many active compounds in addition to inulin. We prepared dry powder using the whole *A. lappa* root rather than just the water-soluble parts, because this better mirrors the actual consumption of *A. lappa* root in cooking.

Previous studies have shown that the short-chain fatty acids (SCFAs) that are produced by the intestinal microbiota during the metabolizing of food reduces caecal pH, leading to an improvement in Ca^2+^ absorption, improvements in the immune system [18], and suppression of the growth of harmful bacteria [19]. SCFAs include acetic acid, butyric acid, propionic acid, formic acid, succinic acid, and valeric acid, and the total concentration of these acids become of the order 100 mM/kg [20]. In the liver, acetic acid is used for energy substrate and fat synthesis, and propionic acid is used in gluconeogenesis. In the intestine, regulatory T cells (Treg) were induced by the stimulation of butyric acid, in addition to immune responses that suppress inflammation, autoimmunity, and allergies [20,21]. One of the transmission pathways between intestinal microbiota and hosts is SCFAs, which are produced by the fermentation of dietary fiber [22,23,24]. Lactic acid, an organic acid, is a precursor of acetic acid and butyric acid. Thus, an increase in the concentration of SCFAs and lactic acid may be a good marker for the effects of inulin and *A. lappa* root on intestinal microbiota.

The circadian clock is the system that regulates the internal daily rhythm. Clock genes such as *Period 1/2*, *Clock*, and *Bmal1* are expressed in the brain and also peripheral tissues [25,26]. The suprachiasmatic nucleus, a part of the brain in the hypothalamus, is known as the central clock, and clocks that are expressed in peripheral tissues are called peripheral clocks. It is known that if circadian clock systems are impaired, many diseases such as metabolic syndrome, heart disease, cancer, and mental disorders can occur [25,27,28,29]. The circadian rhythm can affect the components, localization, and function of the intestinal microbiota [30,31,32,33]. Feeding pattern is an important factor affecting the UniFrac PCoA weighted of microbiota. The presence of *Firmicutes* has a peak during the nighttime diversity of intestinal microbiota. There are many differences between the morning and the evening feeding period and decreases during the daytime fasting period, and *Bacteroides* and *Verrucomicrobia* have peaks during feeding periods [34]. When the timing of feeding was reversed, the cycle of the intestinal microbiota was reversed [30].

Recently, it has been reported that the timing of the intake of functional foods influences their effect. Intake of fish oil containing docosahexaenoic acid and eicosapentaenoic acid early in the active period (breakfast time) caused a stronger effect on the reduction of total blood cholesterol and triglyceride levels in mice fed on a high-fat diet (HFD) compared with intake of fish oil late in the active period (dinner time) [23]. Thus, the circadian clock controls the effects of functional foods through timing of intake. Therefore, in this experiment we examined how the intake timing (breakfast or dinner) of *A. lappa* affected the intestinal microbiota and the concentration of SCFAs.

In the first experiment, we examined whether inulin-rich *A. lappa* root exhibits a similar or different effect on the microbiota of mice compared with pure inulin. In the second experiment, we examined the timing effects of inulin-rich food (morning vs. evening) on the microbiota in a schedule of two meals per day.

## 2. Materials and Methods

### 2.1. Animals

We used eight-week-old male ICR mice (Tokyo Laboratory Animals, Tokyo, Japan). The mice were housed in a room at 22 ± 2 °C and 60 ± 5% relative humidity, with a 12 hour light–dark cycle (lights on 08:00–20:00; lights off 20:00–08:00). The lights-on time was defined as ZT0 and the lights-off time as ZT12. A light intensity of approximately 100 lux was used at the cage level. Throughout all experiments, the mice were able to drink water ad libitum.

### 2.2. Experimental Procedure

The mice were freely fed with a high-fat diet (HFD) (Diet 12451; Research Diets Inc., NJ), which included 45% kcal fat for one week before the experiment started so they became accustomed to the HFD, because it is known that an HFD alters the beta diversity of microbiota [35].

In Experiment 1, an HFD containing cellulose (5.0%) (Oriental Yeast Co., Ltd., Tokyo, Japan), synthesized inulin (1.0, 2.5%) (Fuji FF; Fuji Nihon Seito Co., Tokyo, Japan), or a fine powder of *A. lappa* (1.0, 2.5%) (Mikasa Sangyo Corporation, Yamaguchi, Japan) were fed to mice for ten days under restricted feeding (ZT15–ZT21) (Figure 1A). The predominant molecular size of the synthesized inulin is approximately 2880 [6,7], and it was previously reported that the inulin/fructan size isolated from *A. lappa* root is 2950 [36]. Thus, we used a similar size of inulin/fructan in this experiment.

In Experiments 2 and 3, the mice were fed with *A. lappa* or cellulose, which were mixed in a ratio of 5.0% into the HFD in a schedule of two meals per day; the morning feed was from ZT12 to ZT15 and the evening feed was between ZT21 and ZT24. In Experiment 1, the mice had access to the food for six hours, so in order to equalize the feeding time with this experiment, we defined the morning intake time as the first three hours of the dark period and the evening intake time as the last three hours of the dark period. We gave 5% cellulose or 5% *A. lappa* powder, and access to the food was divided into morning and evening times. The mice were divided into four groups: the ZT19 cellulose group, which ate cellulose in the morning and evening and was sacrificed at ZT19; the ZT4 cellulose group, which ate cellulose in the morning and evening and was sacrificed at ZT4; the ZT19 morning group, which ate *A. lappa* in the morning and cellulose in the evening and was sacrificed at ZT19; and the ZT4 evening group, which ate cellulose in the morning and *A. lappa* in the evening and was sacrificed at ZT4. When the mice were sacrificed, we collected the caecal contents and feces to analyze the SCFA content and the intestinal microbiota (Figure 1B).

All experimental protocols conformed to the laws of the Japanese government and were approved by the Committee for Animal Experimentation at Waseda University (permission #2017-A078).

### 2.3. Measurement of Food Intake Volume

The consumption of food was measured on day 0, day 3, day 7, and day 10. According to the animal ethological assessment, the mice were housed together in groups (*n* = 5). The daily intake of each mouse was calculated by dividing the total amount of food consumed by the number of mice.

### 2.4. Restricted Feeding Machine

In these experiments, we used a restricted feeding machine (KN-680, Natsume Seisakusho Co., Ltd.). The shape of this machine is cylindrical, and half of the side has a net. When the net side goes down, the mice can eat the food inside the net. This machine is regulated by a timer, and the food access time can therefore be controlled.

### 2.5. Measurement of Inulin/Fructan

We analyzed the amount of fructan in the food with a K-FRUC Fructan Assay Kit (Megazyme) according to the manufacturer’s procedure. This kit can measure natural fructan and can also analyze levan-type fructan. If the fructan is a natural fructan, all molecular weights are included in the measurement. However, if the fructan includes hydrolytic fructan, the amount of fructan will be underestimated by approximately 20% because the reducing end of sugars will be reduced to sugar alcohol in the pre-processing process.

First, starch and maltodextrins are hydrolyzed to maltose and maltotriose by pullulanase and β-amylase, and these oligosaccharides are hydrolyzed to d-glucose and d-fructose by maltase (pH 6.5, 40 °C). d-glucose and d-fructose are transformed to D-sorbitol and D-mannitol by alkaline borohydride treatment. In this treatment, natural fructans do not react. Fructan, fructo-oligosaccharides (FOS), and hydrolyzed FOS are hydrolyzed to d-glucose and d-fructose by exo- and endo-inulinase (pH 4.5, 40 °C). d-glucose and d-fructose, which are from fructan, are measured by the 4-hydroxybenzoic acid hydrazide reducing method [37].

### 2.6. Short-Chain Fatty Acid Measurement

SCFAs were measured using gas chromatography and flame ionization detection (Shimadzu Co., Kyoto, Japan), as described by a previous report [38] with some modifications. Firstly, we collected 50 mg of caecal contents to measure SCFAs. We then added the 50 mg contents to 400 μL of dietytl ether (FUJIFILM Wako Pure Chemical Co., Osaka, Japan) and 200 μL of chloroform (FUJIFILM Pure Chemical Co., Osaka, Japan). Thereafter, we added 50 μL of sulfuric acid to extract SCFAs by stirring. Second, we centrifuged the samples at 14,000 rpm for 30 s at room temperature. Then, 1 μL of supernatant was injected into a capillary column (InertCap Pure-WAX (30 m × 0.25 mm, df = 0.5 μm); GL Sciences, Tokyo, Japan). Finally, the supernatant was measured by gas chromatography, first at 80 °C and then at 200 °C. Helium was used as the carrier gas.

### 2.7. Fecal DNA Extraction

According to the experimental method of a previous report [39], the DNA of the intestinal microbiota were extracted from 0.2 g stool samples from each mouse. First, 0.2 g stool were mixed with 20 mL of phosphate-buffered saline, and they were then filtered using a 100 μm nylon mesh (Falcon, LOT: #152668; Corning Inc., USA). Secondly, the filtrate was centrifuged (9000× *g*, 20 min, 4 °C) and pellets were obtained, and these were mixed with 800 μL TE10 buffer and 100 μL lysozyme solution (150 mg/mL) by inversion. After each process, the samples were kept at 37 °C for 1 hour. Third, 20 μL achromopeptidase solution, 50 μL proteinase K, and 20% sodium dodecyl sulfate were added, incubating at 55 °C after each process. Then, the DNA of the intestinal microbiota were extracted by PCI, 3 M sodium acetate, and isopropanol, and refined with 70% ethanol.

### 2.8. 16S rDNA Sequencing

16S rDNA from the intestinal microbiota obtained from the fecal extraction was analyzed by Illumina sequencing according to the 16S Metagenomic Sequencing Library Preparation (15044223 B) protocol.

The V3–V4 variable regions of the 16S rDNA gene were amplified by polymerase chain reaction (PCR) using the following primers:

Forward Primer = 5’-TCGTCGGCAGCGTCAGATGTGTATAAGAGACAGCCTACGGGNGGCWGCAG-3’

Reverse Primer = 5’-GTCTCGTGGGCTCGGAGATGTGTATAAGAGACAGGACTACHVGGGTATCTAATCC-3’

The PCR amplification was carried out with 2.5 μL microbial DNA (5 ng/μL), 5 μL of each primer (1 μmol/L), and 12.5 μL 2×KAPA HiFi HotStart Ready Mix (Kapa Biosystems, Wilmington, MA, USA) in a program of 95 °C for 3 min, 25 cycles of 95 °C for 30 s, 55 °C for 30 s, and 72 °C for 30 s, then 72 °C for 5 min, finally holding at 4 °C. Amplicon PCR products were cleaned using AMPure XP beads (Beckman Coulter Inc., Sacramento, CA, USA) according to the protocol of the report. After obtaining the purified DNA, we completed the index PCR. In this section, the Nextera XT Index Kit v2 (Illumina Inc., San Diego, CA, USA) was used for Illumina sequencing adapters. Index PCR was completed with 5.0 μL PCR production, 5.0 μL each of Nextera XT Index Primer. Then, 25.0 μL 2×KAPA HiFi HotStart Ready Mix and 10.0 μL PCR-grade water were used in a program of 95 °C for 3 min, 8 cycles of 95 °C for 30 s, 55 °C for 30 s, and 72 °C for 30 s, then 72 °C for 5 min, finally holding at 4 °C.

The index PCR products were cleaned using AMPure XP beads (Beckman Coulter Inc., Sacramento, CA, USA) according to the protocol of the report. The quality of the purification was checked using an Agilent 2100 Bioanalyzer with a DNA 1000 Kit (Agilent Technologies, Santa Clara, CA, USA). Finally, the concentration of the DNA library was regulated to 4 nmol/L.

The DNA library was sequenced using the MiSeq Reagent Kit v3 (Illumina Inc.) in the Illumina MiSeq 2 × 300 bp platform. This sequencing was performed according to the manufacturer’s instructions.

### 2.9. Analysis of 16S rDNA Gene Sequencing

The 16S rDNA sequences were analyzed by the Quantitative Insights into Microbial Ecology (QIIME) pipeline version 1.9.1 [40]. The 16S rDNA sequences, which were filtered for quality, were divided into operational taxonomic units (OTU) based on whether they had 97% homology with the UCLUST algorithm [41]. These sequences were then compared to reference sequence collections in the Greengenes database (August 2013 version). A total of 4,901,162 reads were obtained from 80 samples. On average, 61,264.53 ± 4,964.184 sequences were obtained per sample. By using the QIIME pipeline, these sequences were analyzed to produce a taxonomy summary from the phylum to the genus level, the alpha diversity (Simpson diversity index), the beta diversity, and a principal coordinate analysis (PCoA). Principal coordinate analyses were calculated using weighted UniFrac distances.

### 2.10. Predicted Metagenomes

Using the data from experiments 1, 2 and 3 the functional profiles of the microbial communities were predicted by phylogenetic investigation of communities by reconstruction of unobserved states (PICRUSt) [42]. The functional predictions were all assigned to Kyoto Encyclopedia of Genes and Genomes Orthology functional profiles of microbial communities via the 16S sequences. We selected and examined the categories related to “immune system” for analysis simplification and clarification.

### 2.11. Statistical Analysis

All data is shown as the average ± standard error of the mean (SEM) and was analyzed using GraphPad Prism version 6.03 (GraphPad Software). We checked whether this data was normally distributed by the D’Agostino–Pearson test/Kolmogorov–Smirnov test. We also checked for equal variation using Bartlett’s test. After completing these analyses, for the data with one factor, if it showed a normal distribution and equal variation, we carried out a one-way analysis of variance (ANOVA) test, which was analyzed using a parametric test and a Tukey post-hoc test. If the data showed a non-normal distribution or biased variation, we used a nonparametric test; the Kruskal–Wallis test and Dunn’s post-hoc test were used for this. We analyzed the two-factor data using a two-way ANOVA if it was normally distributed and had equal variation. If the data was not normally distributed or did not have equal variation, statistical significance was determined by a Mann–Whitney test or a Kruskal–Wallis test with Dunn’s post-hoc analysis and the two-stage linear step-up procedure of the Benjamini, Krieger, and Yekutieli test for multiple comparisons. A permutational multivariate analysis of variance (PERMANOVA) was used to assess changes in the microbiota composition. The PERMANOVA was analyzed using QIIME.

## 3. Results

### 3.1. Decision of the Appropriate Concentration

To decide the appropriate concentration, we prepared two different concentrations of inulin and *A. lappa* for Experiment 1.

While the inulin and *A. lappa*-containing HFD increased body weight compared to the cellulose group, there were no significant changes of body weight among the groups. The food intake volume of each group was 2.9 g/day/mouse in the cellulose group, 2.8 g in the inulin 1% group, 2.9 g in the inulin 2.5% group, 1.9 g in the *A. lappa* 1% group, and 2.3 g in the *A. lappa* 2.5% group. Thus, the *A. lappa* group showed a reduction in food intake. The inulin intake volumes in the 1% and 2.5% groups were 0.028 and 0.072 g/day/mouse, respectively. The *A. lappa* intake volumes in the 1% and 2.5% groups were 0.019 and 0.058 g/day/mouse, respectively. Considering the comparison between the inulin and the inulin-rich *A. lappa*, we examined the content of inulin/fructan in the *A. lappa*. From the K-FRUC analysis, we found that 50.31 ± 6.04 g (*n* = 6) in the 100 g dry powder *A. lappa* and 0.36 ± 0.02 g (*n* = 6) in the 100 g cellulose powder were estimated to be inulin/fructan.

Under the six-hour feeding condition in the middle of the dark period (Figure 1A), the lactic acid content was significantly higher in the 5.0% cellulose group compared to the 1.0% inulin group and the 1.0% *A. lappa* group. The propionic acid content was significantly higher in the 2.5% inulin group than in the 5.0% cellulose group, the 1.0% inulin group, or the 1.0% *A. lappa* group. The butyric acid content was significantly higher in the 2.5% inulin group than in the 5.0% cellulose or the 1.0% inulin group. Furthermore, the butyric acid content was significantly higher in the 1.0% and 2.5% *A. lappa* groups than in the 5.0% cellulose group. There was a tendency to a dose-dependent increase in the amounts of SCFAs + lactic acid, acetic acid, propionic acid, and butyric acid (Figure 2A).

Since SCFA production was increased, the intestinal microbiota may be affected by the concentration of inulin and *A. lappa*. Therefore, we extracted 16S rDNA from the mouse feces and analyzed the intestinal microbiota. The values of α-diversity using the Simpson index were not significantly different (Figure 2B). The PCoA of weighted UniFrac distances showed that the UniFrac PCoA weighted of the intestinal microbiota composition was significantly different between the 5.0% cellulose group and the 1.0% *A. lappa* group, but not between the 5.0% cellulose group and the 1.0% inulin group. Therefore, there was a significant difference between the 1.0% inulin group and the 1.0% *A. lappa* group. With higher concentrations, there are significant differences between the 5.0% cellulose group and the 2.5% inulin group, and the 5.0% cellulose group and the 2.5% *A. lappa* group (Figure 2C). However, there were no significant differences between the 2.5% inulin group and the 2.5% *A. lappa* group.

As we found clear differences in UniFrac PCoA weighted in the 1% *A. lappa* group compared with the 1% inulin group, changes in the relative abundance of taxonomy were examined in these 1% groups only. At the phylum level, the relative abundance of *Actinobacteria* was significantly increased in the *A. lappa*-intake group compared to the cellulose- and inulin-intake group. Conversely, the relative abundance of *Firmicutes* was significantly decreased in the *A. lappa*-intake group compared to the inulin-intake group (Figure 3A). At the genus level, the relative abundance of *Bifidobacterium* and *Lactobacillus* was increased in the *A. lappa* group compared with the cellulose and inulin groups (Figure 3B). The relative abundance of *Lactococcus* was significantly decreased in the *A. lappa*-intake group compared to the cellulose and inulin-intake groups. The relative abundances of *Streptococcus* and *Oscillospira* were significantly decreased in both the *A. lappa* and inulin groups compared to the cellulose group. The relative abundance of *Staphylococcus* was significantly increased in the inulin-intake group compared to the cellulose and *A. lappa* groups.

In the PICRUSt analysis, which examines the immune system, there were no significant differences among the 1% or 2.5% treated groups (Figure 3C). Regarding the antigen processing and presentation and the nucleotide-binding oligomerization domain-like (NOD-like) receptor signaling pathway, there were significant differences between the 1.0% inulin group and the 1.0% *A. lappa* group (Figure 3D).

These results suggest that low doses of *A. lappa* have strong effects on the microbiota, even though the inulin/fructan content was estimated to be only 50% of the dry *A. lappa* powder.

### 3.2. The Effect of the Timing of A. lappa Intake on the Intestinal Microbiota

In the next experiment, we examined the effects of the timing of *A. lappa* intake on the intestinal microbiota (Figure 1B), because it is known that the circadian rhythm regulates the digestion, absorption, and metabolism of food in the intestine [30].

While morning and evening *A. lappa* intake showed the tendency to increase and decrease body weight, respectively, there were no significant changes in body weight among groups. The food intake volume of each group was 3.5 g/day/mouse in the ZT19 control group, 3.5 g in the ZT19 morning group, 3.9 g in the ZT4 cellulose group, and 3.4 g in the ZT4 evening group. The *A. lappa* intake volumes in the ZT19 morning and ZT4 evening groups were 0.088 and 0.055 g/day/mouse, respectively. The *A. lappa* intake in the evening was similar to the volume in the 2.5% *A. lappa* group (0.058 g/day/mouse) shown in Experiment 1. In comparison with experiment 1 (six hours of feeding during the middle of the dark period), the regime with two separate meals per day (three hours of feeding during the early and late dark period) showed an increase in food intake.

Under the two-meal schedule, the amount of SCFA + lactic acid was significantly higher in the ZT19 morning group than in the ZT4 evening group (Figure 4A). The lactic acid content was significantly increased in the ZT19 morning group compared to the ZT19 control group, and the ZT4 evening group compared to the ZT4 control group. The acetic acid content was significantly higher in the ZT19 morning group compared to the ZT4 evening group. The propionic acid content was significantly higher in the ZT19 morning group compared to the ZT19 control group, and the ZT4 evening group. The butyric acid content was significantly higher in the ZT19 morning group compared to the ZT19 control group, and the ZT4 evening group, and was also higher in the ZT4 evening than the ZT4 control group. The SCFA production tended to increase with *A. lappa* intake (Figure 4A).

Since the SCFA production was increased by *A. lappa* intake, this may indicate that the intestinal microbiota is affected. Therefore, we extracted 16S rDNA from mouse feces and analyzed the intestinal microbiota. The values of the α-diversity using the Simpson index showed no significant differences (Figure 4B). The PCoA of the weighted UniFrac distances showed that the UniFrac PCoA weighted of the intestinal microbiota composition was significantly different between the ZT19 control group and the ZT19 morning group, and between the ZT4 control group and the ZT4 evening group (Figure 4C). From statistical values, control vs. evening (6.57) showed a greater difference when compared with control vs. morning (3.42).

Next, we examined the difference in the changes in the relative abundance of taxonomy. At phylum level, the relative abundance of *Bacteroidetes* was significantly increased in the *A. lappa*-intake group compared to the cellulose-intake group. Conversely, the relative abundance of *Firmicutes* was significantly decreased in the *A. lappa*-intake group compared to cellulose-intake group (Figure 5A). At genus level, the relative abundance of *Rhodococcus* was increased in the ZT19 morning group compared to the ZT4 evening group. The relative abundance of *Bacteroides* was significantly increased in the *A. lappa*-intake group compared to the cellulose-intake group. The relative abundances of *Butyricimonas* and *Ruminococcus* were significantly decreased in the ZT19 morning group compared to the ZT19 control group. The relative abundance of *Streptococcus* was significantly increased in the ZT4 evening group compared to the ZT19 morning group (Figure 5B).

In the PICRUSt analysis, which examines the immune system, there were significant differences between the ZT19 control and the ZT19 morning groups, and between the ZT4 control and the ZT4 evening groups (Figure 5C). Regarding antigen processing and presentation, there were significant differences between the ZT4 control, and ZT4 evening group. Regarding nucleotide-binding oligomerization domain-like (NOD-like) receptor signaling pathway, there were significant differences between ZT19 control and morning group, and between ZT4 control and evening group (Figure 5D).

These results suggest that evening intake of *A. lappa* showed a greater or similarly potent effect on UniFrac PCoA weighted in comparison with morning intake. The amount of *A. lappa* intake of the ZT19 morning group (0.088 g/day for each mouse) was approximately 1.6 times higher than that of the ZT4 evening group (0.055 g/day for each mouse). Therefore, in the next experiment we examined the effect on the intestinal microbiota of lower intake of *A. lappa* in the morning.

### 3.3. The Effect of Smaller Amount of A. lappa Intake in the Morning on the Intestinal Microbiota

In Experiment 2, a small amount of *A. lappa* intake in the evening had a similar effect when compared to a large amount of *A. lappa* intake in the morning. Therefore, in this experiment, we fed mice a smaller amount of *A. lappa* in the morning.

While morning *A. lappa* intake showed a tendency to decrease body weight, there was no significant change in body weight between groups. The food intake volumes of each group were 2.74 g/day/mouse in the cellulose morning group and 3.1 g in the *A. lappa* morning group. The *A. lappa* intake volume in the morning was 0.07 g/day/mouse, and this value was lower than the *A. lappa* intake volume (0.088%) in Experiment 2.

With a smaller volume of *A. lappa* intake in the morning, the butyric acid content was significantly higher in the ZT19 morning group compared to the ZT19 control group. However, lactic acid and other SCFAs, such as acetic acid and propionic acid, were not significantly higher in the ZT19 morning group compared to the ZT19 control group (Figure 6A). There were no significant differences in the total SCFA content + lactic acid.

To examine the intestinal environment, we extracted 16S rDNA from mouse feces and analyzed the intestinal microbiota. The values of α-diversity using the Simpson index did not show a significant difference (Figure 6B). The PCoA of weighted UniFrac distances showed that the UniFrac PCoA weighted of intestinal microbiota composition was not significantly different between the ZT19 control and morning groups (Figure 6C).

Next, we examined the difference in the changes in the relative abundance of taxonomy. At phylum level, the relative abundance of all bacteria showed no significant differences between the ZT19 morning group and the ZT19 control group (Figure 7A). At the genus level, the relative abundances of *Gemella* and *Dorea* were higher in the ZT19 morning group compared to the ZT19 control group. The relative abundance of *Streptococcus* was significantly lower in the ZT19 morning group compared to the ZT19 control group (Figure 7B).

In the PICRUSt analysis, there was no significant difference between the ZT19 control and morning groups (Figure 7C). Regarding antigen processing and presentation and the NOD-like receptor signaling pathway, there were no significant differences between the ZT19 morning and control groups (Figure 7D).

These results demonstrate that when the amount of morning *A. lappa* intake was reduced to be similar to the evening *A. lappa* intake, the effect of the morning *A. lappa* intake on the intestinal microbiota was attenuated. In other words, this indicates that evening *A. lappa* intake has a greater effect on the intestinal environment than morning *A. lappa* intake.

## 4. Discussion

In this study, we chose *A. lappa* root as an inulin/fructan-rich vegetable, because it is a very popular vegetable in Japan. In the first experiment, we compared the effects of pure inulin and *A. lappa* root powder on microbiota diversity under an HFD-induced pattern of microbiota diversity. In this experiment, we used synthesized inulin with a similar molecular size (approximately 3000) of inulin/fructan to that previously noted in *A. lappa* root [36]. In addition, we analyzed the total fructan volume in dry *A. lappa* root powder and found that approximately 50% of the dry weight in *A. lappa* root was inulin/fructan. From these results, we expected 1% and 2.5% *A. lappa* powder to cause a weaker action on microbiota than 1% and 2.5% inulin, respectively. However, surprisingly, *A. lappa* intake resulted in a stronger effect on the UniFrac PCoA weighted of microbiota in feces than inulin intake. *Arctium lappa* root contains various bioactive molecules, such as polyphenols including caffeic acid derivatives and flavonoids, oligosaccharides, polyunsaturated fatty acids, and pectin, a non-inulin polysaccharide [15,17], and it is well known that flavonoids, polyphenols, and also pectin affect the microbiota [43,44]. For example, chlorogenic acid is present in *A. lappa* root in significant quantity, and this bioactive molecule has been shown to modulate gut microbiota in HFD-fed mice [45]. Therefore, it is suggested that *A. lappa* root has a more potent influence on microbiota because of a synergistic and/or additive effect between inulin and the other bioactive molecules. Unfortunately, we could not measure the content of bioactive molecules in the *A. lappa* root powder samples used here. According to a previous report, *A. lappa* contains 19.0 μg/g d.w. of gallic acid, 4388 μg/g d.w. of chlorogenic acid, 25.6 μg/g d.w. of caffeic acid, 7.9 μg/g d.w. of p-coumaric acid, 1447 μg/g d.w. of (-)-epigallocatechin, and 496 μg/g d.w. of (-)-epicatechin [46]. The crude extracts used in the previous report were obtained from ground hade-dried plant material (20 g) in 1:1 ratio of water–ethanol (100:100 mL) at 80 °C for 2 h. Therefore, similar contents of bioactive molecules are expected in the current samples. To conclude a superior effect of *A. lappa* root rather than inulin on gut microbiota, we should measure the contents of such bioactive molecules.

In a previous report, SCFAs were shown to reduce the caecal pH, leading to improved Ca^2+^ absorption, improvements in the immune system [18], and suppression of the growth of harmful bacteria [19]. Propionic acid and butyric acid control the immune function and inflammatory response. According to the previous report, butyric promotes the Treg expression and controls the histone deacetylase (HDAC), which adjusts the negative feedback of gene and NF-κB transcription [47,48]. Furthermore, TNF-α and IL-6 were controlled by SCFAs [49]. In addition, healthy people tend to have more *Bacteroidetes* and fewer *Firmicutes* than obese people [50]. Therefore, it is suggested that the intestinal environment is improved and the immune system is accelerated because of an increase in butyric acid production as a result of *A. lappa* intake. The tentative conclusion of this experiment is that *A. lappa* root may be an ideal vegetable for promoting microbiota diversity, rather than inulin alone.

In the second experiment, we gave a high concentration of *A. lappa* root, such as 5% in the morning or evening under the two-meal schedule, because we wanted to give a similar amount of *A. lappa* at each feeding period. The amount of *A. lappa* was 0.058 g/day/mouse under the six-hour feeding schedule, and 0.055 g/day/mouse during the evening three-hour feeding and 0.088 g/day/mouse during the morning three-hour feeding. With the feeding machine we used, the feeding volume could not be controlled during each access time. While the evening-group mice took a smaller amount of *A. lappa* (63% of that taken by the morning group), both the evening and morning groups exhibited similar increases in SCFA levels and microbiota diversity. These results strongly suggest that intake of *A. lappa* in the evening could be recommended to increase microbiota diversity.

In the last experiment, we reduced the intake of *A. lappa* in the morning by adjusting the concentration of the *A. lappa* powder. In this experiment, the mice ate 0.07 g/day/mouse *A. lappa*, which is still a higher volume of *A. lappa* intake than the evening group. However, there was no significant difference between the cellulose group and the *A. lappa* morning-intake group concerning to the amount of SCFA production and UniFrac PCoA weighted of intestinal microbiota (Figure 6A,C). In the PICRUSt analysis, which relates to the immune system, there was no significant difference between the control group and the *A. lappa* morning-intake group (Figure 7C). Considering antigen processing and presentation, there was no significant difference (Figure 7D). In other words, *A. lappa* intake in the evening has a stronger effect on microbiota than morning intake. However, we do not know the mechanisms for this.

It is known that the metabolic dynamics are different in the morning and the evening, and morning metabolic dynamics are more active than evening metabolic dynamics [51]. Fish oil intake in the morning decreases serum and liver total cholesterol values compared to taking fish oil in the evening in HFD and high-fructose feeding model mice [23]. In addition, food consumption under unsynchronized circadian rhythm is bad for health and increases the risk of disease. In humans and mice, the energy consumption in the early phase of the active period with constant fasting time decreases inflammation, improves the circadian rhythm, increases the resistance to autophagy and stress, and controls the intestinal microbiota [52]. These papers have suggested that morning is a good time for production of a clear reaction. However, intake of catechin-rich green tea in the evening prevents the increase of postprandial glucose levels more effectively than in the morning [53], suggesting the importance of the evening. With *A. lappa* root, the evening intake changed the intestinal microbiota more significantly than morning intake.

One possible mechanism that may explain this is the slow movement of *A. lappa* in the intestine during the inactive period (evening intake) compared with the active period (morning intake). This slow movement may increase the contact time for interaction between the microbiota and the inulin/fructan of *A. lappa*, and this may enhance the effect in the evening. Stress and corticosterone are known to cause dysbiosis of the gut microbiota [54,55], and a high level of glucocorticoid in the morning but not in the evening may be involved in this response. Chlorogenic acid derived from the *A. lappa* causes the increase of diversity [56]. If chlorogenic acid taken in the evening showed a stronger effect, as with catechin-rich tea [53], *A. lappa* root powder may provide a stronger effect on microbiota diversity with evening intake. As the circadian clock affects the bacterial composition of the intestinal microbiota both in the morning and in the evening, evening intake of *A. lappa* root may be combined with morning intake to enhance such bacterial components. These possibilities should be examined in future experiments.

To avoid stress on microbiota induced by isolated individual housing, we housed the mice in groups. Therefore, we could not measure food intake volume for each mouse in the current experiments. There were no differences in body weight in each group (mean ± SE; 42.8 ± 1.3 g for 1% inulin group and 42.3 ± 0.9 g for 1% *A. lappa* group), suggesting a similar amount of food intake for each mouse. However, we should measure food intake in two mice as a group in a future experiment.

Our recent paper demonstrated that SCFAs could reset the phase of the peripheral clock [57]. Therefore, the evening increase in *A. lappa*-root-induced gut SCFAs may cause a delay in the phase of the peripheral clock. Conversely, intake of a relatively high dose of *A. lappa* root in the morning may help to advance the peripheral clock. In general, a morning-type person is known to be healthier than a night-type person. As we took samples at two clock times in the present experiment, circadian expression rhythm of the peripheral clock gene could not be shown. In future experiments, sampling at four to six clock times will be necessary to detect the effect of *A. lappa* on clock gene expression rhythm.

In summary, *A. lappa* root provided a more potent attenuating effect on HFD-induced microbiota diversity change than pure inulin, suggesting that inulin with bioactive molecules from *A. lappa* root has an additive and/or synergic action on microbiota. Intake of *A. lappa* root in the evening caused stronger effect on microbiota diversity change than morning intake. We therefore recommend the habit of evening *A. lappa* root intake as it may help to maintain the health of intestinal microbiota.

## Figures and Tables

**Figure 1 microorganisms-08-00220-f001:**
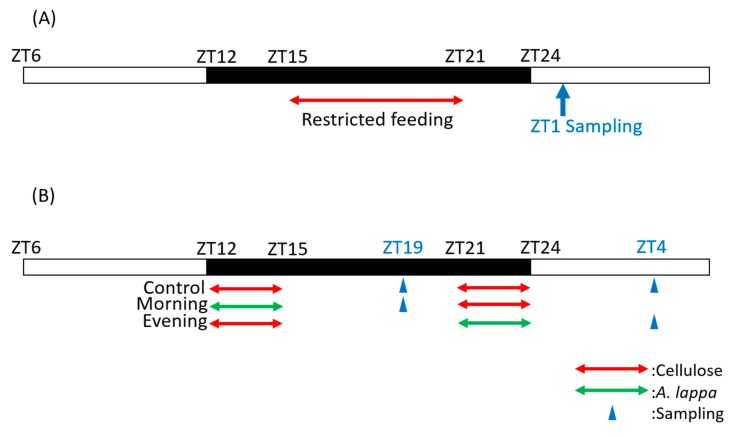
Experimental schedule. (**A**) Scheduling in experiment 1, and (**B**) scheduling in experiments 2 and 3. The white and black bars indicate environmental 12 h light and 12 h dark conditions. The red arrow indicates the feeding time with cellulose and green arrow indicates the feeding time with *A. lappa*. The blue arrow indicates the sampling time. In Experiment 1, mice were fed with each food from ZT15 to ZT21 and sacrificed at ZT1. In Experiments 2 and 3, the mice were fed with each food from ZT12 to ZT15 as morning intake and from ZT21 to ZT24 as evening intake and were sacrificed at ZT19 or ZT4.

**Figure 2 microorganisms-08-00220-f002:**
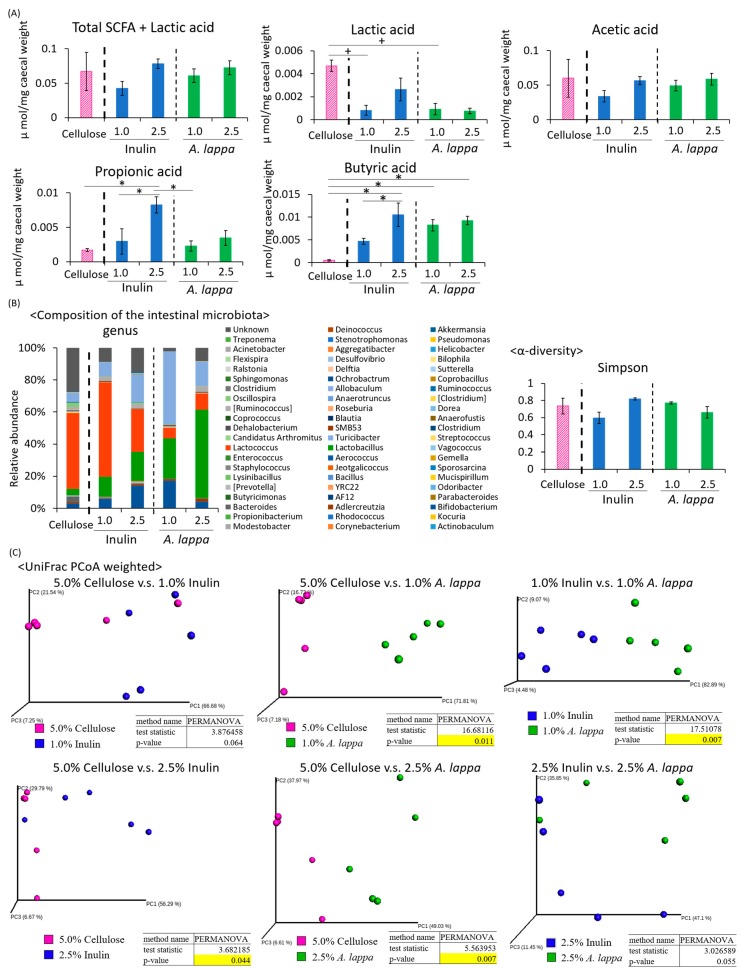
Comparing effect of inulin and *A. lappa* intake on caecal SCFAs and feces microbiota in mice with HFD. Effect of several doses: HFD + 5.0% cellulose, 1.0% inulin, 2.5% inulin, 1.0% *A. lappa*, and 2.5% *A. lappa* on the intestinal microbiota. (**A**) SCFA component and lactic acid was measured from caecal contents by GC-FID. (**B**) Composition of the intestinal microbiota and α-diversity. (**C**) UniFrac principal coordinate analysis (PCoA) weighted intestinal microbiota. All data are represented as mean ± SEM (*n* = 5 for each group). * *p* < 0.05, evaluated using the one-way ANOVA with Tukey post-hoc test. ^+^
*p* < 0.05, evaluated using the one-way ANOVA with a Kruskal–Wallis test and Dunn’s post-hoc test.

**Figure 3 microorganisms-08-00220-f003:**
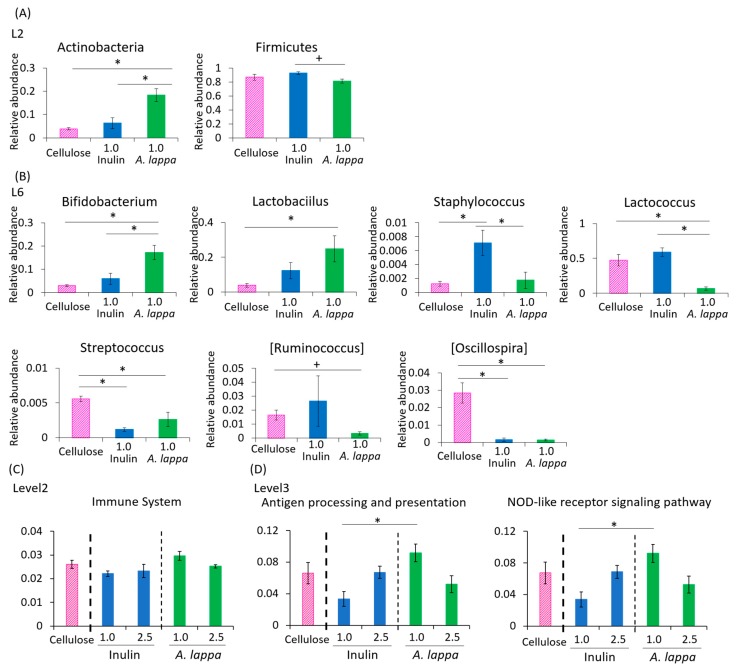
Effect of inulin or burdock on the relative abundance of some bacteria and prediction of function by PICRUSt analysis. (**A**) Relative abundance of phylum level, (**B**) genus level in each group, (**C**) PICRUSt analysis of immune system, and (**D**) antigen processing and presentation and the NOD-like receptor signaling pathway. All data are represented as mean ± SEM (*n* = 5 for each group). * *p* < 0.05 evaluated using one-way ANOVA with Tukey’s post hoc test. ^+^
*p* < 0.05 evaluated using one-way ANOVA with Kruskal–Wallis test and Dunn’s post hoc test.

**Figure 4 microorganisms-08-00220-f004:**
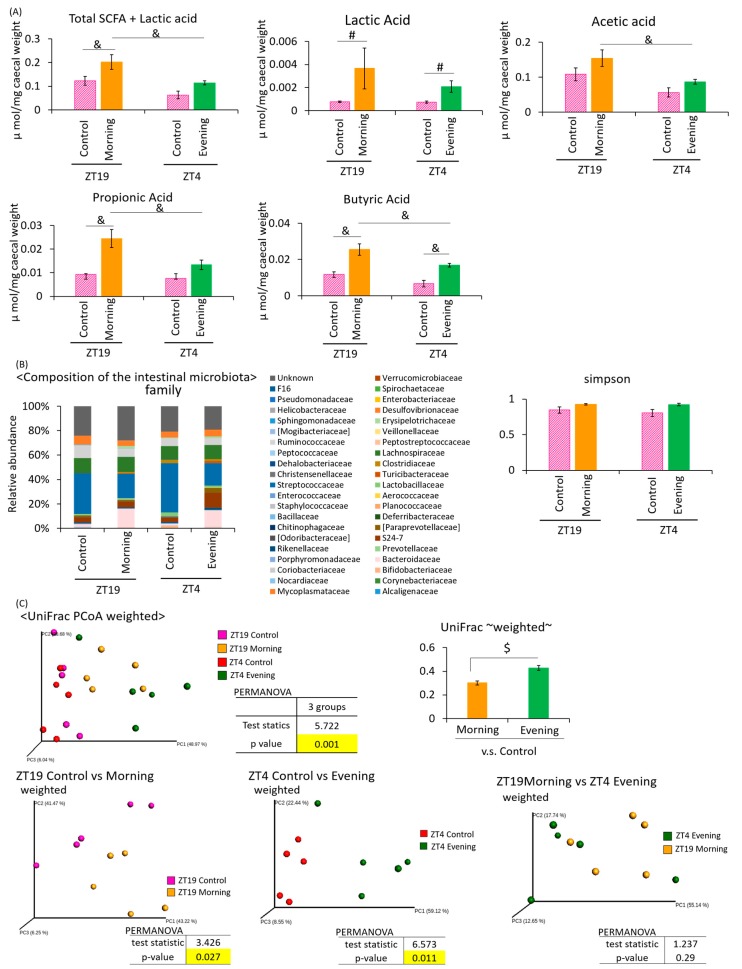
Effect of *A. lappa* intake in the morning or evening on caecal SCFAs and feces microbiota in mice with HFD. Effect of the *A. lappa* intake timing, morning or evening, on the intestinal microbiota. (**A**) SCFA component and lactic acid were measured from caecal contents by GC-FID. (**B**) Composition of the intestinal microbiota and α-diversity. (**C**) UniFrac PCoA weighted of intestinal microbiota. All data are represented as mean ± SEM (*n* = 5 for each group). ^#^
*p* < 0.05, evaluated using the Mann–Whitney test with Dunn’s post-hoc analysis and a two-stage linear step-up procedure of the Benjamini, Krieger, and Yekutieli test for multiple comparisons. ^&^
*p* < 0.05, evaluated using the two-way ANOVA with Tukey post-hoc test. ^$^
*p* < 0.05, evaluated using the t-test. Morning group took *A. lappa* during ZT12–ZT15 and was sacrificed at ZT19. Evening group took *A. lappa* during ZT21–ZT24 and was sacrificed at ZT4.

**Figure 5 microorganisms-08-00220-f005:**
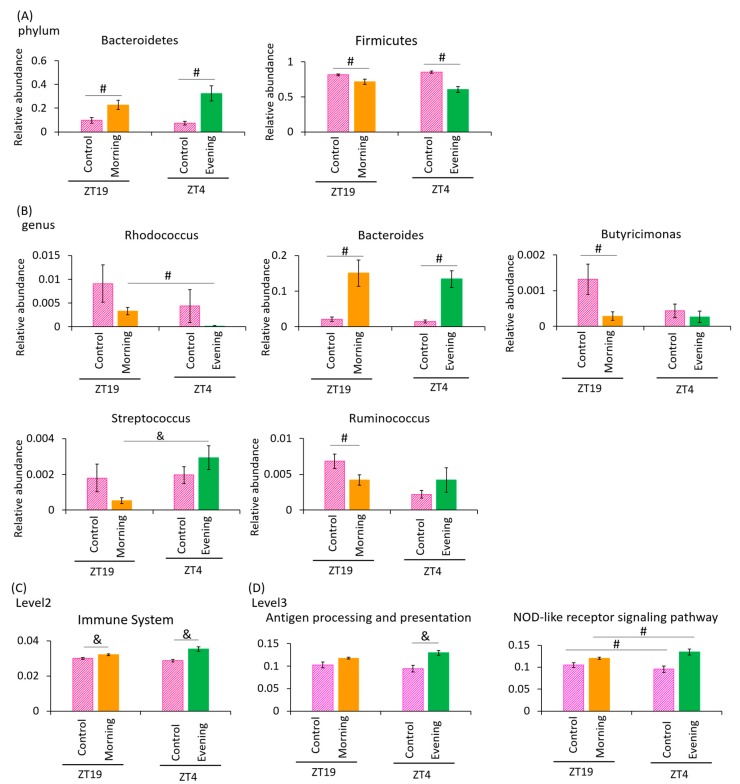
Effect of *A. lappa* intake in the morning or evening on the relative abundance of some bacteria and prediction of function by PICRUSt analysis. (**A**) The relative abundance of phylum level, (**B**) genus level in each group, (**C**) PICRUSt analysis of immune system, and (**D**) antigen processing and presentation and the NOD-like receptor signaling pathway. All data are represented as mean ± SEM (*n* = 5 for each group). ^#^
*p* < 0.05 evaluated using the Mann–Whitney U test with Dunn’s post hoc analysis and two-stage linear step-up procedure of the Benjamini, Krieger, and Yekutieli test for multiple comparisons. ^&^
*p* < 0.05 evaluated using two-way ANOVA with Tukey’s post hoc test. The morning group took *A. lappa* during ZT12–ZT15 and was sacrificed at ZT19. The evening group took *A. lappa* during ZT21–ZT24 and was sacrificed at ZT4.

**Figure 6 microorganisms-08-00220-f006:**
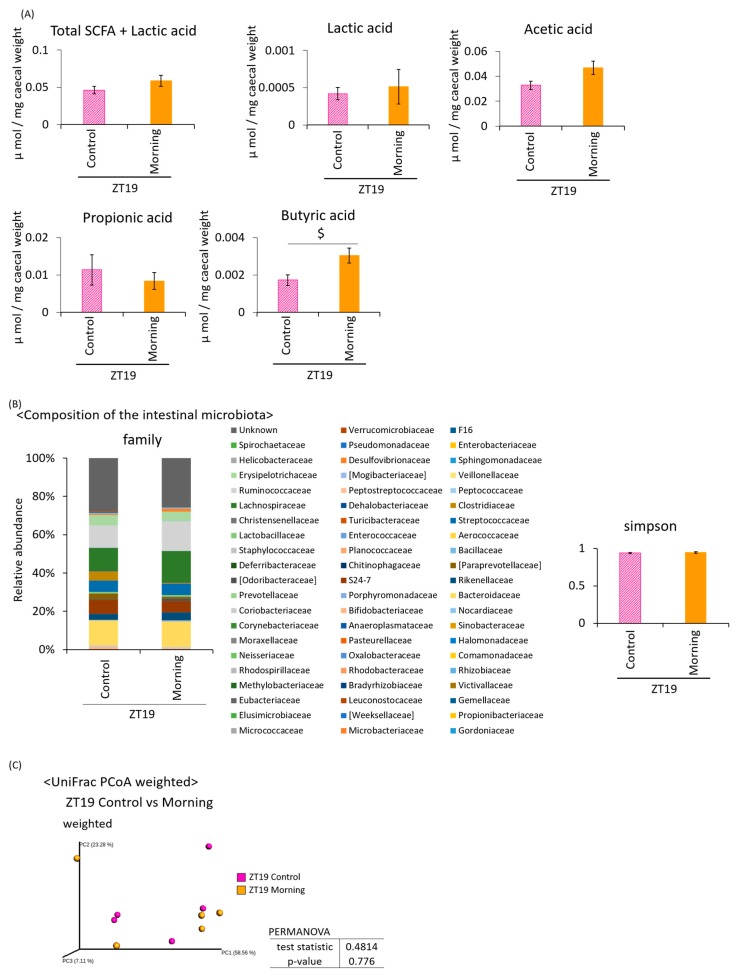
There were no significant differences between the cellulose- and *A. lappa*-intake groups with a small amount of *A. lappa* morning intake Effect of the small volume intake of *A. lappa* in the morning on the intestinal microbiota. (**A**) SCFA component and lactic acid was measured from caecal contents by GC-FID. (**B**) Composition of the intestinal microbiota and α-diversity. (**C**) UniFrac PCoA weighted of intestinal microbiota. All data are represented as mean ± SEM (*n* = 5 for each group). ^$^
*p* < 0.05, evaluated using the t-test. Morning group took *A. lappa* during ZT12–ZT15 and was sacrificed at ZT19.

**Figure 7 microorganisms-08-00220-f007:**
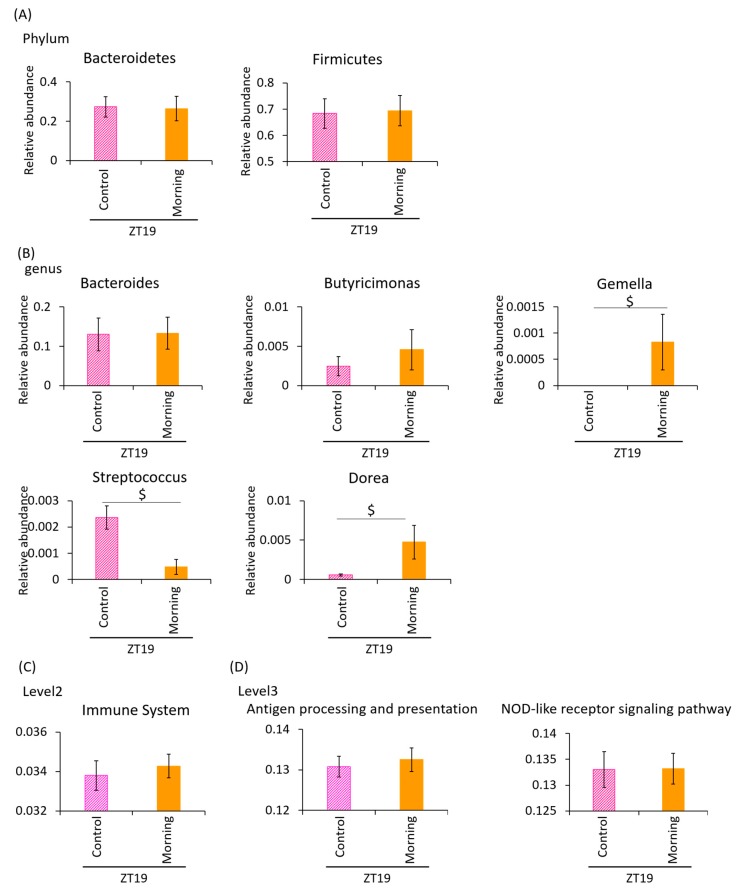
Effect of small-volume intake of *A. lappa* in the morning on the relative abundance of some bacteria and prediction of function by PICRUSt analysis. (**A**) The relative abundance of phylum level, (**B**) genus level in each group, (**C**) PICRUSt analysis of immune system, and (**D**) antigen processing and presentation and the NOD-like receptor signaling pathway. All data are represented as mean ± SEM (*n* = 5 for each group). *p* < 0.05 evaluated using the t-test. The morning group took *A. lappa* during ZT12–ZT15 and was sacrificed at ZT19.

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
