# Peer review of "Effect of Dose and Timing of Burdock (Arctium lappa) Root Intake on Intestinal Microbiota of Mice"

_microorganisms, 2020, doi:10.3390/microorganisms8020220_

Round 1

Reviewer 1 Report

Presented study is very interesting and worth publication however I have some remarks:

Minor remarks

- Arctium lappa should be in Italic, for the first time mention the family – Asteraceae, for chicory and artichoke too. This is of importance as Asteraceae family contains inulin

-Line 46: flavonoids are also polyphenols, please correct the sentence. In Asteraceae family the presence of caffeic acid derivatives is characteristic. Especially, in roots caffeic acid derivatives are more abundant than flavonoids

Major remark

The burdock root ought to be characterized phytochemically (both for inulin content as well as for polyphenols – quality and quantity) otherwise it is difficult to draw conclusion

Author Response

Reviewer 1;

Comments and Suggestions for Authors

Presented study is very interesting and worth publication however I have some remarks:

Minor remarks

- Arctium lappa should be in Italic, for the first time mention the family – Asteraceae, for chicory and artichoke too. This is of importance as Asteraceae family contains inulin

We agree with your suggestion. We have written Arctium lappa in italics.

-Line 46: flavonoids are also polyphenols, please correct the sentence. In Asteraceae family the presence of caffeic acid derivatives is characteristic. Especially, in roots caffeic acid derivatives are more abundant than flavonoids

We have changed the sentence line 50 to “, such as polyphenols including caffeic acid derivatives and flavonoids.”

Major remark

The burdock root ought to be characterized phytochemically (both for inulin content as well as for polyphenols – quality and quantity) otherwise it is difficult to draw conclusion

We have added some comments in the discussion. Please see lines 447-455.

Reviewer 2;

The project is clear even if has been written superficially in some part of it. Anyway, the message of the manuscript is interesting, the results clear and useful for the scientific community. The discussion is appropriate with what is possible to deduce from the experimentation. The design of study is well projected even if it could be better described.

The authors should correct some point of the manuscript:

Raw 59: Lactic acid is not a SCFAs but an organic acid.

We agree with the reviewer’s comment. We have changed the explanation of SCFAs and lactic acid in line 69.

Raw 101: “it is known that an HFD worsens the beta diversity of microbiota”. What did the author mean with “worse”? The beta diversity is a measurement based on comparisons between samples. This index is used just to give an idea of the similarity between the samples.  

We have changed the sentence for accurate expression in line 106.

Raw 126: The daily intake of each mouse was calculated by dividing the total amount of food consumed by the number of mice. This protocol created a big background error. The right calculation of food eaten is fundamental for all the results showed.

We agree  with your suggestion. However, housing the mice in each cage is not ethically appropriate because it would place undue stress on the mice     . This stress would affect their intestinal microbiota. For these reasons, in our current experiment, we evaluated the food consumption under group housing. Interestingly, their body weight on the last day was not different within each group. For example, body weight (mean ± SE) was 42.7 ± 1.3g in the inulin 1% group and 42.3 ± 0.9g in the 1% Arctium lappa group. Thus, we believe that the mice in each housing group      would eat similar amounts of food. We added this argument in the discussion. Please see lines 510-514.

Raw 150: The protocol to be useful needs the concentration of the solution. “… We then added 50 μl of sulfuric acid“ .. is the acid directly added to the 50mg of caecal content? I don’t think so.

We agree with your comment and have corrected the description of the measurement process in lines 158-160.

Raw 159: With the word “genes” the authors mean the DNA. They are different things.

We have made this correction in line 167.

Raw 198: the step “… filtered based on whether they had 97% homology” doesn’t have a sense like this. Did the authors filter the reads based on the 97% homology of the reads with a reference database? Did they use the closed picking python script? The authors should describe better this part to be reproducible.

We have made this correction in lines 207- 209.

I couldn’t succeed in download the raw data of the sequences.

We have attached the data for the sequences.

seq.fnaを添付

I suggest the author to scale the PCoA based on the % of the PCs. Moreover, could be more useful for the reader to put UNIFRAC PCoA weighted instead of the “β-diversity” in the description. Also, the use of purple and red is not the best choice for discriminating the colored points.

PcoA has already been represented by %. We have changed β-diversity to UNIFRAC PcoA weighted in the description. We have changed the colored points from pink to light blue.

Revewer 3;

The manuscript microorganisms-682941 entitled “Effect of dose and timing of burdock (Arctium lappa) root intake on intestinal microbiota of mice” by Watanabe et al have described that Intake of burdock root in the evening had a stronger effect on microbiota diversity in comparison to morning intake. Therefore, it is suggested that habitual consumption of burdock root in the evening may aid the maintenance of healthy intestinal microbiota. I have a few concerns regarding the present manuscript:

Lines 37, 43, 48, 50, among others, please revise the burdock name

We agree with the reviewer’s comments. We have changed the burdock name to A. lappa.

In the introduction section, the authors have stated that there are some important circadian genes, it is a possible measure that genes

We agree with the reviewer’s comments. In this experiment, we sampled only at two clock times. If we examine the effect of cellulose or A. lappa root on clock gene expression, sampling at four clock times or more is necessary. Therefore, we have added some comments in the discussion. Please see lines 519-522.

Some supplementary figures are important in the text, the authors have the intention to move one to the main text?

We have moved some supplementary figures to the text.

Reviewer 2 Report

The project is clear even if has been written superficially in some part of it. Anyway, the message of the manuscript is interesting, the results clear and useful for the scientific community. The discussion is appropriate with what is possible to deduce from the experimentation. The design of study is well projected even if it could be better described.
The authors should correct some point of the manuscript:

Raw 59: Lactic acid is not a SCFAs but an organic acid.

Raw 101: “it is known that an HFD worsens the beta diversity of microbiota”. What did the author mean with “worse”? The beta diversity is a measurement based on comparisons between samples. This index is used just to give an idea of the similarity between the samples.  

Raw 126: The daily intake of each mouse was calculated by dividing the total amount of food consumed by the number of mice. This protocol created a big background error. The right calculation of food eaten is fundamental for all the results showed.

Raw 150: The protocol to be useful needs the concentration of the solution. “… We then added 50 μl of sulfuric acid“ .. is the acid directly added to the 50mg of caecal content? I don’t think so.

Raw 159: With the word “genes” the authors mean the DNA. They are different things.

Raw 198: the step “… filtered based on whether they had 97% homology” doesn’t have a sense like this. Did the authors filter the reads based on the 97% homology of the reads with a reference database? Did they use the closed picking python script? The authors should describe better this part to be reproducible.

I couldn’t succeed in download the raw data of the sequences.

I suggest the author to scale the PCoA based on the % of the PCs. Moreover, could be more useful for the reader to put UNIFRAC PCoA weighted instead of the “β-diversity” in the description. Also, the use of purple and red is not the best choice for discriminating the colored points.

Author Response

Reviewer 2;

The project is clear even if has been written superficially in some part of it. Anyway, the message of the manuscript is interesting, the results clear and useful for the scientific community. The discussion is appropriate with what is possible to deduce from the experimentation. The design of study is well projected even if it could be better described.

The authors should correct some point of the manuscript:

Raw 59: Lactic acid is not a SCFAs but an organic acid.

We agree with the reviewer’s comment. We have changed the explanation of SCFAs and lactic acid in line 69.

Raw 101: “it is known that an HFD worsens the beta diversity of microbiota”. What did the author mean with “worse”? The beta diversity is a measurement based on comparisons between samples. This index is used just to give an idea of the similarity between the samples.  

We have changed the sentence for accurate expression in line 106.

Raw 126: The daily intake of each mouse was calculated by dividing the total amount of food consumed by the number of mice. This protocol created a big background error. The right calculation of food eaten is fundamental for all the results showed.

We agree  with your suggestion. However, housing the mice in each cage is not ethically appropriate because it would place undue stress on the mice     . This stress would affect their intestinal microbiota. For these reasons, in our current experiment, we evaluated the food consumption under group housing. Interestingly, their body weight on the last day was not different within each group. For example, body weight (mean ± SE) was 42.7 ± 1.3g in the inulin 1% group and 42.3 ± 0.9g in the 1% Arctium lappa group. Thus, we believe that the mice in each housing group      would eat similar amounts of food. We added this argument in the discussion. Please see lines 510-514.

Raw 150: The protocol to be useful needs the concentration of the solution. “… We then added 50 μl of sulfuric acid“ .. is the acid directly added to the 50mg of caecal content? I don’t think so.

We agree with your comment and have corrected the description of the measurement process in lines 158-160.

Raw 159: With the word “genes” the authors mean the DNA. They are different things.

We have made this correction in line 167.

Raw 198: the step “… filtered based on whether they had 97% homology” doesn’t have a sense like this. Did the authors filter the reads based on the 97% homology of the reads with a reference database? Did they use the closed picking python script? The authors should describe better this part to be reproducible.

We have made this correction in lines 207- 209.

I couldn’t succeed in download the raw data of the sequences.

We have attached the data for the sequences. However, you need QIIME software which is available on WEB site.

I suggest the author to scale the PCoA based on the % of the PCs. Moreover, could be more useful for the reader to put UNIFRAC PCoA weighted instead of the “β-diversity” in the description. Also, the use of purple and red is not the best choice for discriminating the colored points.

PcoA has already been represented by %. We have changed β-diversity to UNIFRAC PcoA weighted in the description. We have changed the colored points from pink to light blue.

Reviewer 3 Report

The manuscript microorganisms-682941 entitled “Effect of dose and timing of burdock (Arctium lappa) root intake on intestinal microbiota of mice” by Watanabe et al have described that Intake of burdock root in the evening had a stronger effect on microbiota diversity in comparison to morning intake. Therefore, it is suggested that habitual consumption of burdock root in the evening may aid the maintenance of healthy intestinal microbiota. I have a few concerns regarding the present manuscript:

Lines 37, 43, 48, 50, among others, please revise the burdock name

In the introduction section, the authors have stated that there are some important circadian genes, it is a possible measure that genes

Some supplementary figures are important in the text, the authors have the intention to move one to the main text?

Author Response

Reviewer 3;

The manuscript microorganisms-682941 entitled “Effect of dose and timing of burdock (Arctium lappa) root intake on intestinal microbiota of mice” by Watanabe et al have described that Intake of burdock root in the evening had a stronger effect on microbiota diversity in comparison to morning intake. Therefore, it is suggested that habitual consumption of burdock root in the evening may aid the maintenance of healthy intestinal microbiota. I have a few concerns regarding the present manuscript:

Lines 37, 43, 48, 50, among others, please revise the burdock name

We agree with the reviewer’s comments. We have changed the burdock name to A. lappa.

In the introduction section, the authors have stated that there are some important circadian genes, it is a possible measure that genes

We agree with the reviewer’s comments. In this experiment, we sampled only at two clock times. If we examine the effect of cellulose or A. lappa root on clock gene expression, sampling at four clock times or more is necessary. Therefore, we have added some comments in the discussion. Please see lines 519-522.

Some supplementary figures are important in the text, the authors have the intention to move one to the main text?

We have moved some supplementary figures to the text.
